# Effect of Annealing on the Interface and Properties of Pd/Al Composite Wires

**DOI:** 10.3390/ma16041545

**Published:** 2023-02-13

**Authors:** Jiabin Gui, Zhen Yang, Xiangqian Yin, Haofeng Xie, Lijun Peng, Wenjing Zhang, Xujun Mi

**Affiliations:** 1State Key Laboratory of Nonferrous Metals and Processes, China GRINM Group Co., Ltd., Beijing 100088, China; 2GRIMAT Engineering Institute Co., Ltd., Beijing 101407, China; 3General Research Institute for Nonferrous Metals, Beijing 100088, China

**Keywords:** Pd/Al composite wires, annealing, interface, intermetallic compounds

## Abstract

This paper investigates the changes in the interface organization and properties of 0.10 mm Pd/Al composite wires annealed at different temperatures. The optimum comprehensive performance of the material was obtained after annealing at 300 °C for 120 s. Its tensile strength, conductivity and elongation are 140.61 MPa, 46.82%IACS and 14.89%, respectively. Scanning electron microscopy (SEM) and transmission electron microscopy (TEM) were used to observe the intermetallic compounds on the interface. The annealing temperature and the formation heat of intermetallic compounds determine the categories and evolution of intermetallic compounds. When the thickness of the intermetallic layer is more than 1 μm, it has a serious effect on the electrical conductivity and elongation of the materials.

## 1. Introduction

Bi-metallic composite wire [1] is a composite wire with a core and an outer layer made up of two different metals or alloys. Bimetallic composite wire has excellent properties of two materials, can achieve the performance that a single material cannot, and is widely used in all walks of life. For example: copper-clad aluminium [2,3,4,5] composite wire has both the good electrical conductivity of copper and the light weight and good ductility of aluminium, and is widely used in high frequency transmission cables and other fields. Silver-clad aluminium [6,7] has excellent electrical conductivity, good contact properties and low density, and is used as aerospace wire and waveguides. Pd/Al composite wire [8] is an aluminium-based composite bimetallic wire with the advantages of low density, light weight and the good electrical conductivity of aluminium. As a reactive thermal fuse, it can be supplied with energy from AC, DC or capacitive discharge to heat the composite wire. As the rate of Pd/Al diffusion increases, a large number of intermetallic compounds are generated at the interface, and an immediate exothermic reaction occurs, generating high temperatures that can directly detonate the explosive or ignite the solid fuel. Because of its short reaction time, the availability of a constant current to control the reaction process and the high heat release during the reaction (about 1368.19 J/g), it can directly meet the requirements of short ignition response time, small ignition peaks and easy control of the ignition time for solid rocket motors [9,10,11,12]. It has the characteristics of having less substance discharged during the reaction, heat transferred mainly by conduction, the reaction does not need oxygen, and it can react under vacuum and inert gas. It is suitable for detonation underwater, underground, at high altitude and in a vacuum. It is used as a detonating wire in fields such as torpedoes and new electronic detonators [13,14].

The ignition reaction of Pd/Al composite wires mainly depends on the reaction heat during the formation of intermetallic compounds. However, the production of intermetallic compounds not only affects the basic mechanical and electrical properties of the composite wire, but also plays a key role in the effectiveness of the application of the composite wire. According to the Al-Pd phase diagram [15], it can be seen that the Pd/Al phase diagram is complex, with the existence of nine different equilibrium phases depending on the elemental content, namely Al_4_Pd, Al_3_Pd, Al_21_Pd_8_, Al_3_Pd_2_, AlPd, Al_3_Pd_5_, Al_2_Pd_5_ and AlPd_2_ phases. Many researchers have carried out correlation analyses for the performance of Pd/Al composite wires at different temperatures [16,17] and ignition responses [18]. Howard [19] found that PdAl_3_ was the initial phase of Pd/Al composites at 250 °C and 300 °C. The presence of the Al_3_Pd_2_, AlPd, Al_3_Pd_5_ and AlPd_2_ phases was observed in Pd/Al composites heat treated at 200 °C to 500 °C by KÖster and Hon [20]. The presence of PdAl_3_ and PdAl_4_ phases was also observed. It is believed that β-AlPd is the first to be formed in all samples and is accompanied by the formation of the Al_3_Pd_2_ phase. Nastasi and Hung [21] found PdAl_4_ as the initial phase by in situ annealing, and with further heat treatment, the PdAl_4_ and PdAl_3_ phases were obtained. However, these studies rely on scanning electron microscopy, energy spectral analysis, X-ray diffraction, etc., and cannot make a precise determination of the crystal structure of the phases.

According to the Cu-Al diffusion couple [22,23], after holding at a certain temperature for a period of time, a series of different intermetallic compounds will appear on the interface. The heat treatment system is the main factor affecting the bonding of Cu-Al. However, the growth of intermetallic compounds in Pd/Al composites after annealing at different temperatures remains unclear. Therefore, in order to ensure that the Pd/Al composites obtained meet the application requirements, in the preparation and processing of composite materials, it is necessary to have an in-depth understanding of the types of compounds on the interface and the sequence of compounds generated under the specific preparation process, so as to better control them.

In this study, a combination of transmission electron microscopy and scanning electron microscopy was used to observe the intermetallic compound layer changes of Pd/Al composite wires after annealing. The aim is to investigate the generation of phases during annealing on the Pd/Al interface, and to investigate the effect of the intermetallic compounds on the overall properties of Pd/Al composite wires. This work seeks a deeper understanding of the phase formation and transformation, the relationship between different intermetallic compounds and the properties of Pd/Al composite wires.

## 2. Experimental Methods

Palladium tubes with an outer diameter of 32.00 mm and an inner diameter of 25.80 mm and solid aluminium rods with an outer diameter of 25.80 mm were prepared using 99.999% pure palladium and pure aluminium in a volume ratio of 35:65. After drawing, the Pd/Al composite wire diameter (D) was 0.10 mm.

Next came annealing of the 0.10 mm Pd/Al composite wire at 200 °C, 250 °C, 300 °C and 350 °C for 5 s, 10 s, 30 s, 60 s, 120 s, 300 s, 900 s, 1800 s, 3600 s, 7200 s and 10,800 s, respectively. The TH-8203 S tensile testing machine was used to carry out the tensile test, in which the length of the gauge was 200 mm and the tensile speed was 50 mm/min. The conductivity was measured by TX-300 conductivity tester (Xiamen, China). In order to ensure the accuracy and reliability of the data, five sets of data were tested for each sample in each regime, and the results were averaged. The data of the sample in its initial state without heat treatment were also tested for comparison.

The annealed samples were inlaid using a cold inlay on a cross-section of palladium-clad aluminium wire, after coarse grinding, fine grinding and polishing. The intermetallic compounds at the material interface were observed by scanning electron microscopy (SEM) and transmission electron microscopy (TEM), where TEM samples were obtained by focused ion beam (FIB) preparation. The component content was analysed by an energy dispersive spectrometer (EDS). The thickness of the intermetallic layer was measured by energy spectrum results.

## 3. Results and Discussion

### 3.1. Interface Analysis

Figure 1 shows an SEM image of the cross-section of an unheated treated Pd/Al composite wire. At low magnification (as shown in Figure 1a), it can be seen that the material has a clean cross-section with no impurities or oxides present (The outer white part is Pd and the inner black part is Al). Due to the different hardness of Pd and Al, the Pd/Al interface shows a cogwheel shape after intense cold deformation. Under high magnification (as shown in Figure 1b), it can be seen that the interface is tightly bonded and no holes exist, indicating that the interface of the Pd/Al composite wire prepared by the cold-drawing method is well-bonded.

Figure 2 shows SEM images of Pd/Al composite wires annealed at 200 °C for different times, with Al in the darker areas and Pd in the lighter areas due to the difference in lining degree. As can be seen from the figure, when the annealing time is relatively short (as shown in Figure 2a), no intermetallic compound layer is observed at the interface of the material. When the annealing time reached 60 s (as shown in Figure 2b), a thin layer of the intermetallic compound appeared at the interface of the two phases of the material. With a further increase in annealing time, the intermetallic compound layer of the material further thickened, increasing to an average of about 0.85 μm at 7200 s, and when the annealing reached 10,800 s (as shown in Figure 2d), the thickness of the interfacial layer reached about 1.1 μm.

Figure 3 shows TEM images of Pd/Al composite wires annealed at 200 °C for 5 s and 10,800 s. Figure 3a,e show high-angle annular dark-field (HAADF) diagrams for 5 s and 10,800 s annealed at 200 °C. It can be seen that at an annealing time of 5 s, a very thin intermetallic compound layer appears at the interface. Through high-resolution images according to fast Fourier transform (FFT), this was identified as a hexagonal Al_3_Pd_2_ phase in the (0 2 −1) direction (as shown in Figure 3c). Position 4 in Figure 3f shows the polycrystalline ring, according to the selected area electron diffraction (SAED) result, confirmed as a hexagonal Al_3_Pd_2_ structure. Figure 3g shows a selected diffraction electron image at position 5. Position 5 reveals a hexagonal Al_3_Pd_2_ phase in the (1 −1 2) direction (black quadrilateral in Figure 3g) and a tetragonal Al_21_Pd_8_ (red quadrilateral in Figure 3g) phase in the (1 −1 0) direction. This indicates that at 200 °C, the material interface is dominated by the growth of the Al_3_Pd_2_ phase and the formation of a small portion of the Al_21_Pd_8_ phase.

Figure 4 shows SEM images of Pd/Al composite wires annealed at 250 °C for different times. A thin interfacial layer can be seen at 250 °C when the annealing time reaches 30 s (as shown in Figure 4b) at the interface of the material. With increasing annealing time, the intermetallic compound layer at the interface of the material thickens rapidly with an average thickness of about 0.67 μm when the annealing time is increased to 1800 s (as shown in Figure 4c). It is worth noting that with a further increase in annealing time, the thickness of the interfacial layer of the material does not vary significantly, with annealing time of 3600 s (as shown in Figure 4d) and 10,800 s (as shown in Figure 4e), respectively, both being around 1.163 μm. It indicates that, in the early stages of annealing, the mutual diffusion of Pd and Al is rapid due to direct contact, but as the intermetallic compound layer is created, the diffusion drive decreases, resulting in the slow growth of the interfacial layer. This phenomenon was also seen in the samples annealed at 200 °C.

Figure 5 shows TEM images of annealing at 250 °C for 5 s and 10,800 s. Figure 5a shows that, at this temperature, the material generates intermetallic compounds on the interface at the early stage of annealing that are extremely small and discontinuous, with an intermetallic compound thickness of about 30 nm, and the generated intermetallic compound is a hexagonal Al_3_Pd_2_ phase in the (0 0 1) direction (as shown in Figure 5c). At an annealing time of 10,800 s, two different intermetallic compound layers appeared at the interface of the material, in the region of position 2 and position 3 in Figure 5e, with the hexagonal Al_3_Pd_2_ phase (black quadrilateral in Figure 5f,g) and the tetragonal Al_21_Pd_8_ phase (red quadrilateral in). According to the diffusion phase ratio, only one phase can be present in the diffusion region for both metals. The EDS results suggest that the region at position 2 is the Al_3_Pd_2_ phase region and position 3 is the Al_21_Pd_8_ phase region. The presence of the Al_21_Pd_8_ phase was not directly observed at the beginning of the annealing process, indicating that the growth of the Al_21_Pd_8_ phase was backward. Due to the thin thickness of the Al_21_Pd_8_ phase, it was not observed in Figure 4.

Figure 6 shows SEM images of the Pd/Al composite wire annealed at 300 °C for different times. When the annealing time reaches 30 s (as shown in Figure 6b), a grey intermetallic compound layer appears at the Pd/Al interface bond (red arrow area in the figure). With a further increase in annealing time, by 900 s (as shown in Figure 6c), the presence of the intermetallic compound layer at the interface of the material is clearly visible at the interface, where the thickness of this compound layer is approximately 0.52 μm. When the annealing time reached 10,800 s (as shown in Figure 6f), the interfacial layer of the material was further thickened to approximately 3.15 μm. The higher diffusion drive allows further growth of the interfacial layer as the temperature increases compared to 250 °C.

Figure 7 shows TEM images of annealing at 300 °C for 5 s and 10,800 s. It can be seen that, at an annealing time of 5 s, a discontinuous intermetallic compound layer appears at the interface, which, after EDS calibration and FFT transformation at high resolution, is identified as a hexagonal Al_3_Pd_2_ phase in the (0 1 0) direction, with an intermetallic compound layer thickness of approximately 20 nm. At an annealing time of 10,800 s, it was observed that the intermetallic compound layer of the material was only one layer of the material, consisting of a large number of fine grains, and the material showed a polycrystalline ring structure, which was considered to be a hexagonal Al_3_Pd_2_ structure by calibration.

Figure 8 shows SEM images of Pd/Al composite wires annealed at 350 °C for different times. It can be seen that, at an annealing time of 5 s (as shown in Figure 8a), a white continuous intermetallic compound layer is observed at the interface of the material, and as the annealing time increases by 60 s (as shown in Figure 8a), the intermetallic compound layer of the material grows to about 0.48 μm. At an annealing time of 300 s (as shown in Figure 8d), the thickness of the interfacial layer of the material increased to 1.01 μm. This indicates that the interfacial layer grew faster and faster as the annealing temperature increased. When the annealing time came to 3600 s (as shown in Figure 8e), an off-white layer appeared at the interface, and it is believed that a new intermetallic compound was produced at this time. At an annealing time of 10,800 s (as shown in Figure 8f), the new intermetallic compound grows and appears as a white area in the diagram.

Figure 9 shows TEM images of annealing at 350 °C for 5 s and 10,800 s. At an annealing time of 5 s, a continuous and homogeneous layer of the intermetallic compound appears at the interface of the material with an interfacial layer thickness of approximately 7.3 nm. Based on the EDS results at position 1 (Figure 9f) and its Fourier transform image (shown in Figure 9c), it is considered that an orthogonal Al_3_Pd_5_ phase in the (1 0 −2) direction is generated at this point. As the annealing time was increased to 10,800 s (shown in Figure 9e), two compound layers appeared at the interface of the material, where the thin layer showed crescent-shaped growth, and the combination of EDS results (shown in Figure 9h) and SAED results showed that the intermetallic compound layers at positions 2 and 3 were the Al_3_Pd_2_ and AlPd layers, respectively.

The interfacial intermetallic compounds generated by annealing the material under different conditions are shown in Table 1. An intermetallic compound Al_3_Pd_2_ phase was formed at the interface at the beginning of annealing at 200–300 °C. At 350 °C, the Al_3_Pd_5_ phase was formed. However, the Al_3_Pd_5_ phase is a high temperature phase that is stable above 615 °C. Traces of the Al_3_Pd_5_ phase were only observed above 600 °C by A.S. Ramos [16]. A similar situation also appears in Cu/Al composite wires [24]. This is due to the fact that Al is in full contact with Pd and the reaction diffuses at a higher temperature, resulting in a higher instantaneous heat release and, thus, a very high temperature. On the other hand, due to the high solid solution of Al in Pd and the very low solid solution of Pd in Al, it is easier for Al to diffuse into Pd and preferentially reach a concentration suitable for the formation of the Al_3_Pd_5_ phase. As the annealing time increases, the diffusion of atoms slows down, the temperature decreases and stabilises and the Al_3_Pd_5_ phase transforms into the Al_3_Pd_2_ and AlPd phases. After prolonged annealing, the complex effects of external heat transfer, exothermic generation of pre-metallic compounds and the diffusion rates of Al and Pd lead to the generation of different phases at the interfaces annealed at different temperatures. After annealing at 200 °C, 250 °C and 300 °C for 10,800 s, the intermetallic compound at the interface is the Al_3_Pd_2_ and Al_21_Pd_8_ phases, and after annealing at 350 °C for 10,800 s, the intermetallic compound at the interface is the Al_3_Pd_2_ phase.

### 3.2. Mechanical and Conductive Properties

Figure 10 shows the variation of tensile strength, elongation and electrical conductivity of the Pd/Al composite wires at different temperatures with annealing time. The mechanical and electrical properties of the materials are seriously affected by intermetallic compound layers [19,20]. It can be seen that the initial tensile strength, elongation and electrical conductivity of the materials are 269.91 MPa, 1.22% and 48.08% IACS. When the annealing temperature is 200 °C, the tensile strength and electrical conductivity of the composite wire show a smooth and then decreasing trend, while the elongation shows a smooth and then increasing trend. The inflection point occurs in the time range of 3600 s–7200 s. The reduction in tensile strength and the increase in elongation are mainly due to the recovery of Pd and the recrystallisation of Al. The decrease in electrical conductivity, on the other hand, is due to the growth of interfacial intermetallic compounds.

As the annealing temperature increases, the annealing time used for the inflection point to occur becomes shorter, and the decrease in tensile strength and electrical conductivity becomes more significant. The elongation of the composite wire reaches its peak at 200 °C annealed for 3600 s, 250 °C annealed for 1800 s, 300 °C annealed for 120 s and 350 °C annealed for 120 s, and the tensile strength and electrical conductivity remain at a high level when the thickness of the interfacial layer of the material is all around 1 μm. As the annealing time increases further and the thickness of the interfacial layer continues to increase, the electrical conductivity and elongation of the material decreases sharply.

There is a similar effect of intermetallic compound layers at the copper-clad aluminium interface on material elongation [25,26,27]. This is due to the tendency of the thick intermetallic compound layer to break up during stretching [28,29], and to fail to co-ordinate the simultaneous deformation of Pd/Al, leading to interface separation and severely reducing the elongation of the wires. It can therefore be assumed that 1 μm is the critical point at the interface where the intermetallic compound layer affects the performance of the composite wire. The overall properties of the material were optimal when the material was annealed at 300 °C for 120 s, at which point the tensile strength, elongation and electrical conductivity were 140.61 MPa, 14.89% and 46.82% IACS, respectively.

The conductivity of the material was 48.28% IACS and 45.86% IACS when annealed at 300 °C for 5 s and 350 °C for 5 s, respectively, at which time the interfacial phases were the Al_3_Pd_2_ and Al_3_Pd_5_ phases, but the thickness of the Al_3_Pd_2_ phase (20 nm) was less than that of the Al_3_Pd_5_ phase (7.8 nm). Considering the better recovery of Pd when annealed at 350 °C, it suggests that the negative effect of the Al_3_Pd_5_ phase on the conductivity is greater than that of the Al_3_Pd_2_ phase.

## 4. Conclusions

It was shown in this study that the intermetallic compound on the interface of the Pd/Al composite wire is in the Al_3_Pd_2_ phase at the beginning of annealing at 200–300 °C. At 350 °C, a high temperature stable Al_3_Pd_5_ phase is generated. Due to the full contact between Al and Pd, reaction-diffusion occurs at a higher temperature, and the instantaneous heat release is large, thus reaching a very high temperature and generating an Al_3_Pd_5_ phase. The elongation of the material is severely reduced when the thickness of the interfacial layer exceeds 1 μm. The optimum performance of the material was obtained at 300 °C after annealing at 300 °C for 120 s. Its tensile strength, conductivity and elongation are 140.61 MPa, 46.82%IACS and 14.89%, respectively.

## Figures and Tables

**Figure 1 materials-16-01545-f001:**
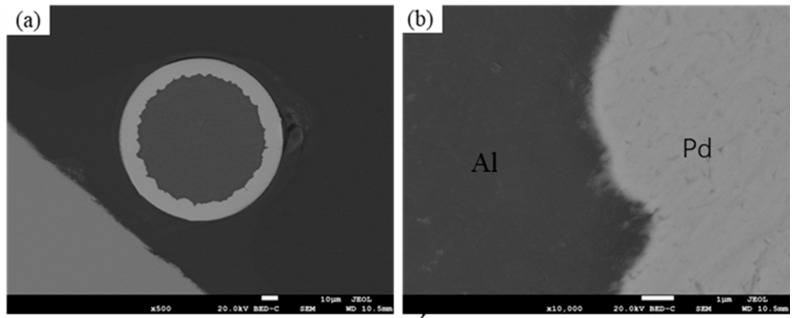
SEM images of Pd/Al composite wire: (**a**) ×500, (**b**) ×10,000.

**Figure 2 materials-16-01545-f002:**
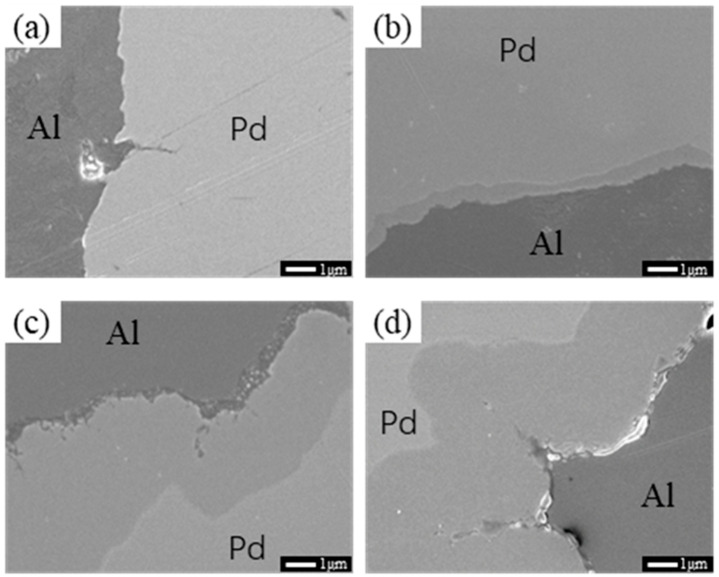
SEM images of Pd/Al composite wires annealed at 200 °C for different times; (**a**) 5 s, (**b**) 60 s, (**c**) 7200 s, and (**d**) 10,800 s.

**Figure 3 materials-16-01545-f003:**
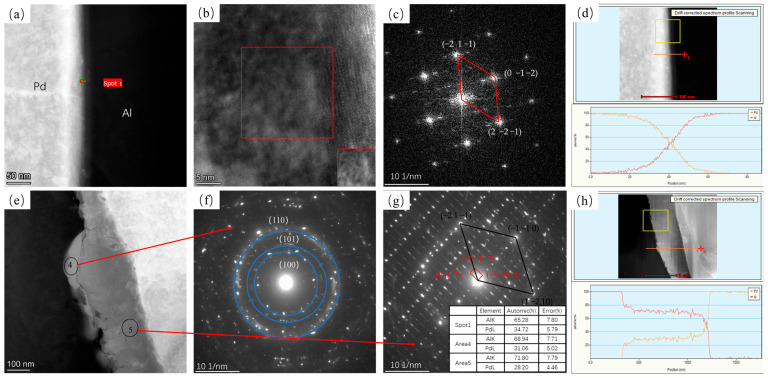
TEM plots of Pd/Al composite wires annealed at 200 °C for different times and their SEAD plots; (**a**) HAADF plot at 5 s, (**b**) high-resolution plot of spot 1 in (**a**,**c**) FFT transform of the red region in (**b**,**d**) EDS line sweep at 5 s, (**e**) HAADF plot at 1080 s, (**f**) SAED plot of spot 4 in (**e**,**g**) SAED plot of spot 5 in (**e**) and EDS results for each position, and (**h**) EDS line sweep at 10,800 s.

**Figure 4 materials-16-01545-f004:**
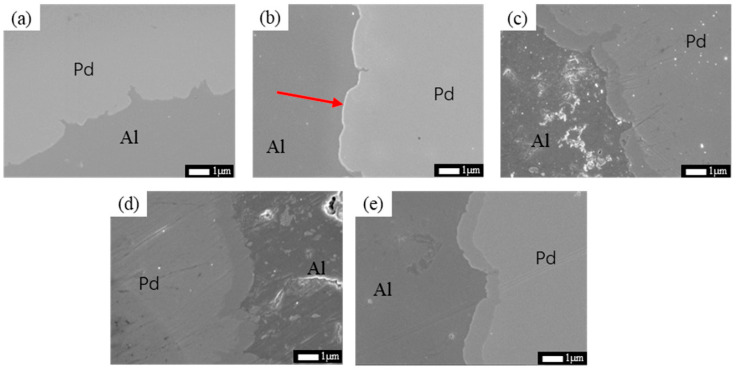
SEM images of Pd/Al composite wires annealed at 250 °C for different times; (**a**) 5 s, (**b**) 30 s, (**c**) 1800 s, (**d**) 3600 s, and (**e**) 10,800 s.

**Figure 5 materials-16-01545-f005:**
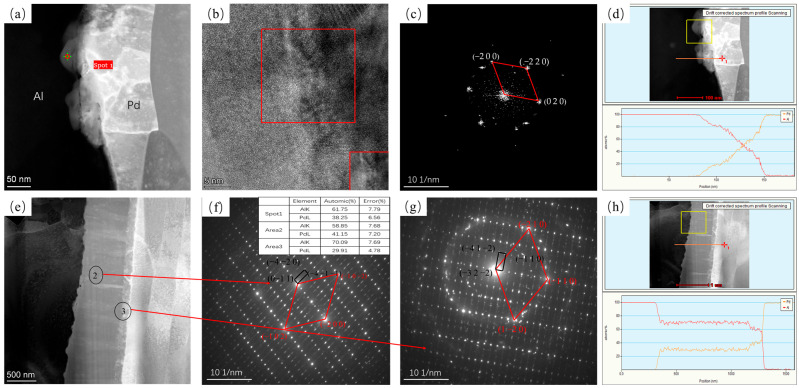
TEM plots of Pd/Al composite wires annealed at 250 °C for different times and their SEAD plots: (**a**) HAADF plot at 5 s, (**b**) high-resolution plot of spot 1 in (**a**,**c**) FFT transform of the red region in (**b**,**d**) EDS line sweep at 5 s, (**e**) HAADF plot at 1080 s, (**f**) SAED plot of spot 2 in (**e**,**g**) SAED plot of spot 3 in (**e**) and EDS results for each position, and (**h**) EDS line sweep at 10,800 s.

**Figure 6 materials-16-01545-f006:**
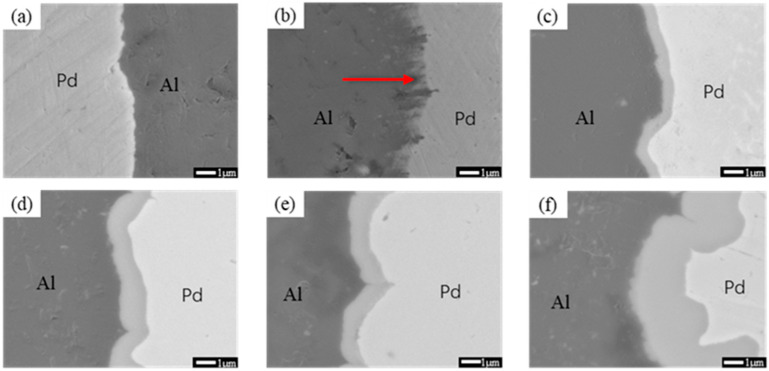
SEM images of Pd/Al composite wires annealed at 300 °C for different times; (**a**) 5 s, (**b**) 60 s, (**c**) 900 s, (**d**) 3600 s, (**e**) 7200 s, and (**f**) 10,800 s.

**Figure 7 materials-16-01545-f007:**
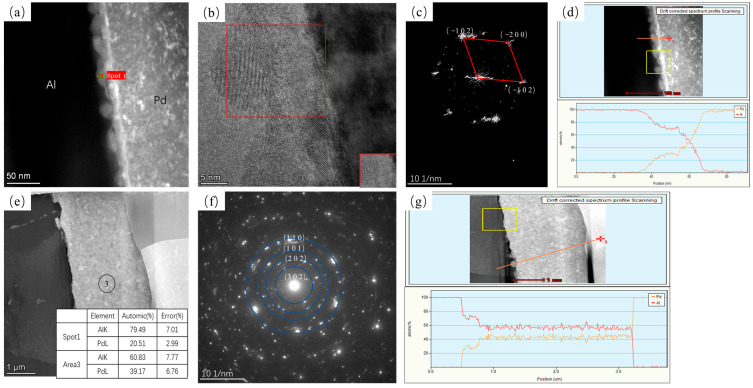
TEM plots of Pd/Al composite wires annealed at 300 °C for different times and their SEAD plots. (**a**) HAADF plot at 5 s, (**b**) high-resolution plot of spot 1 in (**a**,**c**) FFT transform of the red region in (**b**,**d**) EDS line sweep at 5 s, (**e**) HAADF plot at 1080 s and EDS results for each position, and (**f**) SAED plot of spot 3 in (**e**,**g**) EDS line sweep at 10,800 s.

**Figure 8 materials-16-01545-f008:**
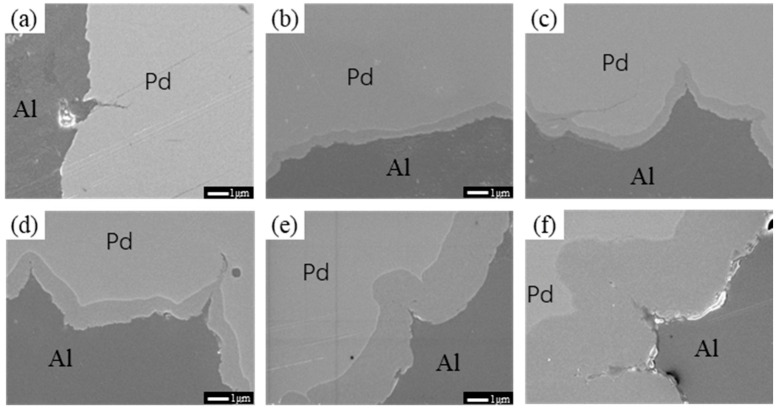
SEM images of Pd/Al composite wires annealed at 350 °C for different times; (**a**) 5 s, (**b**) 60, (**c**) 120 s, (**d**) 300 s, (**e**) 3600 s, and (**f**) 10,800 s.

**Figure 9 materials-16-01545-f009:**
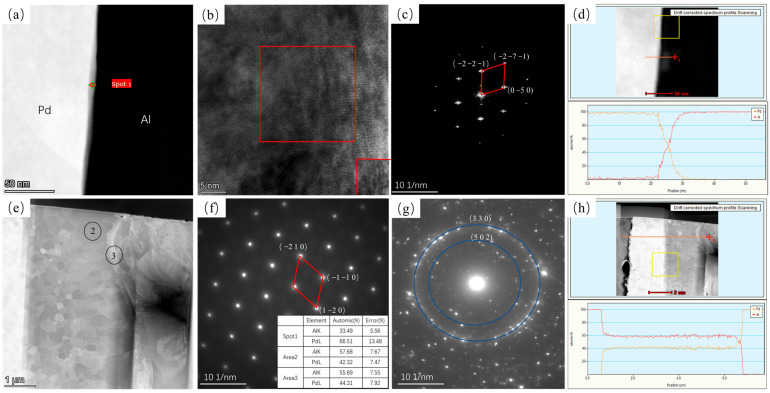
TEM plots of Pd/Al composite wires annealed at 350 °C for different times and their SEAD plots. (**a**) HAADF plot at 5 s, (**b**) high-resolution plot of spot 1 in (**a**,**c**) FFT transform of the red region in (**b**,**d**) EDS line sweep at 5 s, (**e**) HAADF plot at 1080 s, (**f**) SAED plot of spot 2 in (**e**,**g**) SAED plot of spot 3 in (**e**) and EDS results for each position, and (**h**) EDS line sweep at 10,800 s.

**Figure 10 materials-16-01545-f010:**
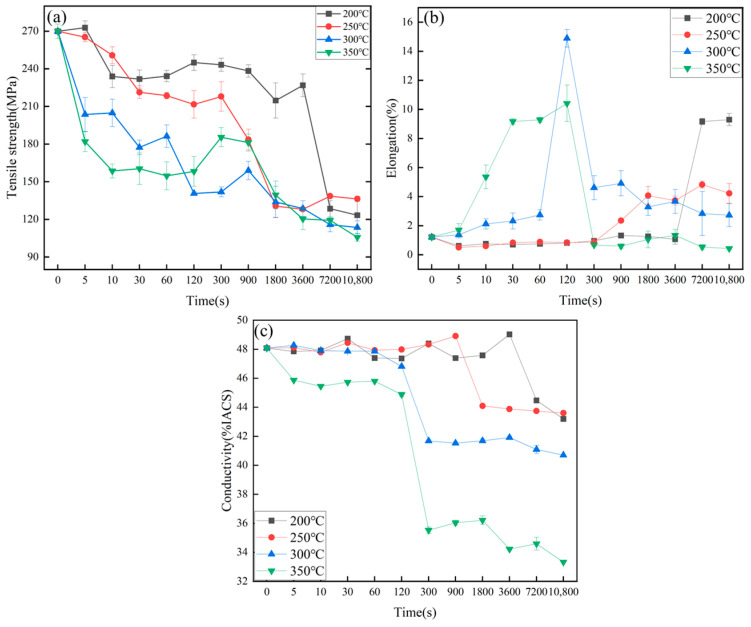
Properties of Pd/Al composite wires at different temperatures with annealing time: (**a**) tensile strength, (**b**) elongation, and (**c**) electrical conductivity.

**Table 1 materials-16-01545-t001:** Interfacial intermetallic compounds generated under different annealing conditions for Pd/Al composite wires.

T(°C)	Time(s)	
	5	10,800
200	Al_3_Pd_2_	Al_3_Pd_2_, Al_21_Pd_8_
250	Al_3_Pd_2_	Al_3_Pd_2_, Al_21_Pd_8_
300	Al_3_Pd_2_	Al_3_Pd_2_
350	Al_3_Pd_5_	Al_3_Pd_2_, AlPd

## Data Availability

The raw data cannot be shared at this time as the data are also part of an ongoing study.

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
