# Peer review of "Effect of Annealing on the Interface and Properties of Pd/Al Composite Wires"

_materials, 2023, doi:10.3390/ma16041545_

Round 1

Reviewer 1 Report

The manuscript ID "materials-2175825" having the title "Effect of annealing on the interface and properties of Pd/Al composite wires " has described the studies related to the annealing temperature effects on Pd/Al composite. This is not good, and it should be extensively revised in order to be acceptable for materials. 

My specific comments on this paper are as follow:

  1. Abstract of the manuscript is poorly written; authors should improve the abstract. Graphical abstract should be provided.
  2. Authors are trying to publish a thesis work as a paper, which is good, but extensive proofreading is very much needed. Please check line no: 68 page number 3.
  3. I don’t know whether thesis is placed online in their institute or not? but if it is a thesis; will authors provide assurance that there will be no copyright issues from institution?
  4. Please mention elements in the figure 2, which part is Al and which part is Pd. Same for figure 3 b , c, and d and figure 7 and figure 9.
  5. Please check for abbreviation mistakes. Some are not explained. HAADF? What is SEAD? Please explain the full form at least.
  1. Conclusion section should not be bulleted. Please write conclusion in a laconic way.

7.      I have seen many syntax errors; there are errors in punctuation marks in the whole manuscript. Some sentences are incomplete, making it hard to understand what do authors want to say? Authors should read and revise the manuscript for grammatical errors too. Overall, writing is not coherent and not up to the mark.

Reviewer 2 Report

The manuscript reports the results on an experimental investigation of the effect of annealing temperature and time on the interface characteristics and properties of Pd/Al composite wires. The paper has major weaknesses as follow.

1.      The main weakness of the paper is the very limited meaningfulness and significance of the study. The novelty of the work should be clearly justified.

2.      The English of manuscrpt needs a major revision. Some sentences are impossible to understand. The manuscript is prepared with least amount of attention and contains numerous typo mistakes. I believed that paper MUST be proofreaded before any review process.

3.      Referencing style in the manuscript and also in reference section is not accurate and consistent. It needs to be checked to be according to journal guideline.

4.      There are not proper citation and references to support the findings.

In more details:

Line 13: What do you mean by macroscopic properties?

Line 16: what is histomorphology and what is its application in your study. Please elaborate.

Line 22: “External heat transfer……”. Please limit the abstract to the main and more significant finding of your study not discussion which are not related to our study.

Line 26: First sentence is not clear at all.

Line 31: “… is quoted …?

Line 40: pillar? What is it?

Line 41 “… reaction (about 327 calories per gram)”. Can you explain?

Line 57: “Howard[16] et al. found”. Please check the in-text referencing style. It should follow the journal guideline. Check all references especially in Introduction section.

Line 63: “however, these studies…..”. what the point behind the sentence?

Line 67: Figure 1 is not necessary. Please delete it.

Line 68: In this thesis?

Line 69: “….interface organization…..”? What do you mean?

Line 70: “….order of phase generation…”?. Please explain.

Line 75: 32mm or 32 mm?

Line 78: “… with a diameter D of 0.10 mm…”. Diameter (D) of 0.10 mm.

Line 81: “The electrical…”. The sentence is not clear.

Line 88: “…observed and calibrated…”. What do you mean by calibrated?

Line 89: Define the acronyms throughout the abstract and introduction such as SEM, TEM and FIB. Also define the meaning of anyone of them when they are first mentioned.

Line 94: Fig.2a or Fig. 2a?

Line 94: “…material has a clean cross-section with no foreign matter present…”. Please elaborate what foreign matter?

Line 99: Casing drawing method?

Line 100: Please highlight different areas in Figure 1. It implies in all Figures. Highlight intermetallic layer in Figure 3.

Line 101: Using consistent form; Figure or Fig

Line 110: “……reached about 1.1μm.”. How did you measure the thichkness of interfacial layer? Please explain in experimental method.

Line 114: Please explain the method of obtaining chemical composition in the experimental method.

Line 114: EDS line scan results are not clear. Provide them in separate figure.

Line 121: “…layer of transition zone…” what do you mean?

Line 122: “….Diffraction spot calibration….” What is it?

Line 142: What Red arrow shows in Fig. 5b.

Line 150: “…were calibrated…”.?

Line 164: “…a layer of grey 164 diffusion zone…”. Please elaborate.

Line 165: What is difference between interface layer, transition zone, intermetallic compund layer. There is many terms used. Are those the same? If yes, use a consistent term. If not, please explain.

Line 173: “ ….. discontinuous jagged intermetallic compound layer….”.?

Line 218: The sentence need references. There are also other statements that does not come from the original results and then need references. It is clear that there is only 1 reference in the results and discussion which does not support your findings.

Line 245: “….initial tensile strength….”.What do you mean by initial values? What is your reference sample? Non-heat treated sample? Please explain it in the experimental method section.

Line 249: “The reduction in tensile strength….”. Need reference.

Line 252: Please elaborate why the electrical conductivity increases with growth of intermetallic layer? Provide reference.

Line 258: “…..material is all around 1μm”. Better determine what samples, what temperature and annealing time.

Line 263: “….reducing the elongation of the filaments”. Need reference. Plus explain wire or filament?

Line 271: Please explain how to measure the thickness of the Al3Pd2 phase to be 20 nm?

Line 299: Referencing style is not consistent.

Reviewer 3 Report

1. In Figure 2, the arrows should indicate the layers corresponding to palladium and aluminum.

2. In Figures 1-9, the scale (label) is poorly distinguished.

3. Figure 4 f – EDA data is poorly readable. Most likely, the authors presented the EDA data in “atomic” percentages, and not in “automic"! A similar remark is made for Figure 6e, 8d, 10e.

4. Figure 8f is of poor quality, poorly readable.

5.            Line 218 - “This is due to the fact that Al is in full contact with Pd and the reaction diffuses at a higher temperature, resulting in a higher instantaneous heat release and thus a very high temperature”. What kind of reaction is in question, what kind of product is formed, at what temperature? How does this result correlate with the Pt/Al state diagram?

6. How does the annealing time affect the thickness of the reaction zone between palladium and aluminum?

7. What samples were used for the tense test?

8. It is necessary to improve English.

Round 2

Reviewer 1 Report

Comments has been addressed

Reviewer 2 Report

All comments are addressed